# Enhanced YOLOv8-based pavement crack detection: A high-precision approach

**ZuXuan Zhang[1], HongLi Zhang[1]\*, TongJia Zhang[2]**

**1** School of Engineering Machinery, Shandong Jiaotong University, Jinan, China, **2** School of Mechanical and Electrical Engineering, Shandong Jianzhu University, Jinan, China

\* 63027776@qq.com

## Abstract

At present, the repair of cracks is still implemented manually, which has the problems of low identification efficiency and high labor cost. Crack detection is the key to realize the mechanical and intelligent crack repair. To solve these problems, an improved automatic recognition algorithm based on YOLOv8 model, YOLOV8-DGS is proposed in this study. Firstly, this paper introduces deep separable Convolution (DWConv) into YOLOv8 backbone network to capture crack information more flexibly and improve the recognition accuracy of the model. Secondly, GSConv is used in the neck part to reduce computation and enhance feature representation, especially in the processing of multi-scale fracture features. Through these improvements, YOLOv8-DGS not only improves the accuracy of small cracks, but also ensures the real-time and high efficiency of intelligent joint filling equipment in practical applications. Experimental results show that the Precision, Recall, F1-score, mAP50 and FPS of the YOLOv8-DGS algorithm in pavement crack detection are 91.6%, 90%, 90.8%, 92.4% and 85 frames, respectively. At the same time, the recognition rate of different types of cracks in the model reached more than 86%, which increased by 20.5% compared with the YOLO11 model. This method can provide theoretical basis for automatic crack identification and technical support for automatic seam filling machine.

## 1. Introduction

Pavement crack is one of the key factors affecting road safety and life. With the acceleration of urbanization, the problem of pavement damage has become increasingly serious, especially the appearance of cracks, which not only affects road safety, but also accelerates the aging of pavement [1]. The traditional crack detection method usually relies on manual inspection, and the road inspection personnel observe the crack on the highway with the naked eye, generally the crack detection takes 2–3 hours per kilometer. This method is time-consuming and laborious, and the detection of the road section with heavy traffic is extremely dangerous, prone to the problem of missing detection, and the detection effect is not ideal. In recent years,

**Data availability statement:** The experimental dataset that support the findings of this study are openly available in [GitHub] at [https://github.com/sekilab/RoadDamageDetector]

**Funding:** Natural Science Foundation project of Shandong Province Project number: ZR2024QE374 Project name: Research on the key technology of robot six-degree-of-freedom grasping and detecting of highly reflective parts in unordered stacking scene The funders had no role in study design, data collection and analysis, decision to publish, or preparation of the manuscript.

**Competing interests:** The authors have declared that no competing interests exist.

with the rapid development of computer vision and deep learning technology, the pavement crack detection method based on deep learning has gradually become a research hotspot, and has been widely used in intelligent engineering equipment [2].

At present, YOLO (You Only Look Once) series models have made remarkable achievements in the field of target detection because of their high detection speed and accuracy. YOLOv8, as the classic version of the YOLO series, is widely used in object detection. However, in practical applications, the precision and accuracy of YOLOv8's small target detection in complex scenes are still not enough to meet the needs of intelligent devices. Specifically, the current YOLOv8 model is still difficult to achieve high enough accuracy when dealing with small cracks, irregularly shaped cracks, and partially obscured cracks. In addition, the existing pavement crack detection methods such as CrackFormer mostly focus on the detection task itself, but how to further improve the robustness and practicality of the algorithm is still a technical problem to be solved when conducting real-time and accurate crack detection for intelligent crack filling equipment.

In order to realize automatic crack repair work, it is necessary to get rid of relying on manual detection of road diseases, this method is low efficiency, subject to the subjective influence of maintenance personnel, prone to misdiagnosis and missed diagnosis [3]. This study summarizes the previous research results and further breakthroughs in computer vision. Cai et al. [4,5] used YOLACT network model to identify ice images in the cold ocean, improved the detection accuracy of ice floes in images captured in the complex polar environment, and used improved image processing algorithms to estimate the circumference crack size of the identified ice. Liu et al. [6] proposed a Crack Transformer network (CrackFormer) for fine-grained crack detection to optimize the common problems of uneven strength, complex topology, low contrast and noise background in crack images. Although the problems such as background are solved, there are still misdetection and missing detection of small targets. With the rapid development of deep learning, crack detection based on image processing is faced with more complex challenges [7]. Jia et al. [8] used the improved VarifocalNet model to analyze and detect photovoltaic module defects, and designed two new bottleneck modules to enhance the network depth and receptive field, thereby improving the speed and accuracy of defect detection. However, traditional convolutional neural networks still have some limitations when dealing with complex recognition tasks, and deep learning techniques are constantly optimized to further improve the recognition accuracy of the models. Aiming at the problems such as poor recognition and classification effect, slow recognition speed and weak generalization ability of current recognition methods, Jiang et al. [9] proposed an algorithm based on improved YOLOv8, which integrates CloFormer module to improve the ability to extract high and low frequency features from Marine debris images and improve the overall performance of the model. In the study of hidden crack location of asphalt pavement, Zhen et al. [10] added Resnet50vd to YOLOv3 for improvement. The improved convolution is a hyperparameter optimization method based on Bayesian search, which enhances the detection capability of the model. In addition, Liu et al. [11] also proposed a YOLOv3 model containing four-scale detection layers, which

effectively reduces the missed detection rate of crack characteristics of small targets. It is worth noting that image quality is closely related to the accuracy and speed of model recognition. Sun et al. [12] propose an image enhancement algorithm for SLAM, which aims to solve the problem of significant decline in target detection accuracy in vision degradation scenes, and significantly improve the detection performance of mainstream target detection networks such as YOLOv3, Faster R-CNN and DetectoRS. In view of the shortcomings of deep learning in data acquisition and defect statistics in crack detection, Cai et al. [13] proposed a borehole rescue command and decision system for underground disaster areas based on multi-source heterogeneous data fusion. The indicators of the human pose fusion image recognition algorithm of the system are higher than those of traditional algorithms. Recent studies have demonstrated the effectiveness of multi-source heterogeneous data fusion in real-time identification of critical situations in disaster areas. Xu et al. [14] designed a three-dimensional reconstruction and geometric topography analysis method for small lunar craters, and obtained three-dimensional point clouds on the lunar surface by means of the aggregation of stereo matching networks. Through the test of KITTI2015 dataset, the advantages of this system in terms of real-time performance and effectiveness were proved.

Convolutional neural network is more suitable for complex and small target detection such as road cracks [15]. This study aims to improve the YOLOv8 model and propose a high-precision crack identification method suitable for intelligent seam filling machine based on the characteristics of road crack detection. In order to reduce the computational complexity and improve the inference speed of the model, the convolutional modules in the backbone and neck parts are optimized. Instead, the more lightweight deep separable convolutional modules and the grouping shuffle convolutional modules are replaced. Before training, image filtering and histogram equalization algorithm are introduced to improve the robustness of the model, so as to meet the real-time detection requirements of intelligent sewing machine. The goal of this study is to improve the accuracy and accuracy of crack detection to 91.6% by improving the YOLOv8 model while maintaining high speed, so as to provide technical support for the automatic operation of intelligent sewing machine.

## 2. Materials and methods

### 2.1. Datasets

In this study, the road crack pictures collected by the author named RDD2022 are used for training and testing. The RDD2022 dataset contains road defect data from six countries: China, Japan, Czech Republic, Norway, USA and India. In this study, high-quality pictures in RDD were manually screened and the local pavement crack pictures were combined to make a data set. The data set was divided into categories by running codes. There were four labels in the data set, namely D00 (longitudinal crack), D10 (transverse crack), D20 (block crack) and D30 (irregular crack). A rich and balanced data set of pavement cracks suitable for this study is obtained.

According to Table 1, a total of 4878 pavement cracks were selected in this data set, among which 4104 types of cracks were D00; D10 total 2359; D20 total 934; Total number of D40 is 321. First, these images are cropped to a resolution of 640×640 to fit the training parameters. Then the data is expanded by rotation, translation and other operations of the image, and the image is expanded to 19512. Finally, in order to eliminate the noise and blur in the image, the filtering equalization operation is used to further enhance the data. Data partitioning is a crucial step in the development of large

Table 1. Classification of pavement cracks.

| Type | Feature code | Label | Number |
|---|---|---|---|
| Longitudinal crack | D00 | C | 4104 |
| Transverse crack | D10 | L | 2359 |
| Block crack | D20 | B | 934 |
| Irregular crack | D30 | I | 321 |

models, and common training set to test set ratios are 9:1, 8:2, 7:3. Although a high proportion of training sets will improve the learning ability of the model, the training results will be unstable due to insufficient test sample size, and can not truly reflect the generalization ability of the model. Although a high proportion of test sets will improve the generalization ability of the model, it will lead to insufficient model learning on some features, and it can not fully learn the features of the data. Therefore, the ratio of 8:2 is adopted in this data set to ensure sufficient model training data and stable model evaluation results, which is not prone to large fluctuations. Through the above operations, a road crack data set covering different lighting conditions, barrier-free occlusion and different material pavement is produced, and the model trained by this data set can be better generalized to different scenes.

The rectangular boxes with different colors in Fig 1(a)-(d) are the characteristic boxes of longitudinal crack type, transverse crack type, block crack type and irregular crack type respectively. labelimg software [16] was used for annotation. The causes of cracks can be roughly divided into four aspects: pavement material characteristics, climate conditions, traffic load and design standards, so that the pavement is subjected to the action of transverse/longitudinal forces to produce transverse/longitudinal cracks; The transverse cracks extend horizontally, perpendicular to the center line of the road, and the spacing is uniform. Longitudinal cracks extend longitudinally, parallel to the center line of the road, and have local branches. Block crack is a kind of criss-cross crack form, which is typical of asphalt pavement block crack. When longitudinal cracks and transverse cracks continue to develop, it is easy to form massive cracks. Temperature fatigue, reflection cracks and asphalt aging can also cause massive cracks [17].

## 2.2. Image Preprocessing

### 2.2.1. Histogram equalization.
The histogram of the image represents the frequency of different gray levels (brightness values) in the image. The goal of histogram equalization [18] is to make the histogram distribution of the image more uniform by redistributing the gray level of the image, so as to improve the contrast of the image. Each image is composed of individual pixels, in order to solve the histogram equalization of the image, it is necessary to use a discrete

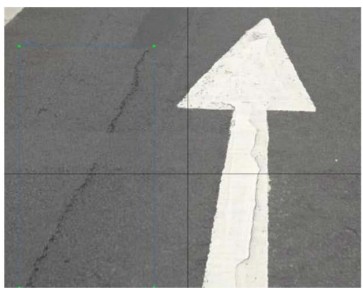
(a) Longitudinal cracks

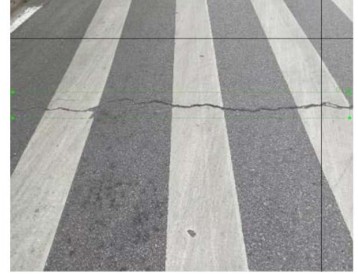
(b) Lateral cracks

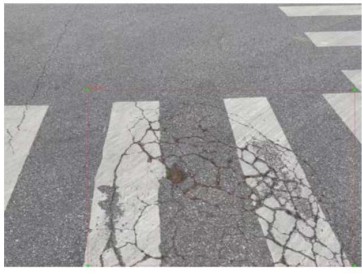
(c) Massive cracks

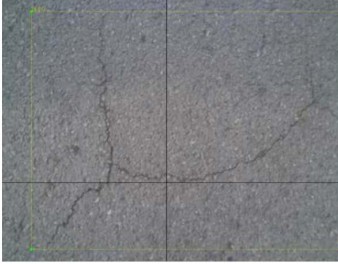
(d) Irregular cracks

**Fig 1. Sample label.**

form of cumulative distribution function, which is a function representing the cumulative value of the occurrence frequency of each gray level and all previous gray levels. Its calculation formula is as follows:

$$CDF(i) = \sum_{j=0}^{i} p(j)$$
(1)

Where, $p(j)$ is the probability density of the gray level $j$, $i$ is the value of the gray level.

In order to map the range of the cumulative distribution function to [0, 255], normalization is usually performed:

$$CDFnorm(i) = \frac{CDF(i) - CDF\min}{N \times M - CDF\min}$$
(2)

Where, $CDF_{min}$ is the minimum value of the cumulative distribution function, N and M are the number of rows and columns of the image, respectively. According to the normalized $CDF$ value, the gray value of the original image is mapped to the new gray value, and the enhanced image is generated.

The traditional histogram equalization method may cause detail loss or noise for the image with good contrast, which will affect the image quality.

**2.2.2. Filter equalization enhancement.** In order to solve the problem of noise amplification and excessive enhancement caused by histogram equalization in image enhancement, a filter equalization enhancement algorithm is designed in this paper based on the application requirements of intelligent sewing machine. The method calculates the contrast of the image for many times. When the contrast exceeds the default threshold, the image is directly output. If the contrast is lower than the default threshold, the histogram equalization operation to limit the contrast is further applied to effectively avoid excessive enhancement in the traditional equalization method.

Fig 2 shows the flow of filtering equalization enhancement algorithm. The initial contrast threshold is set as T=1.8 in the experiment. When the road crack image is taken as input, bilateral filtering is first applied, and the spatial distance parameter is 4, the spatial arrangement parameter and the color space parameter are 75 and 80, respectively, to obtain the filtered image. Then, the contrast value between the filtered image and the original image is calculated. If contrast T<1.8, histogram equalization of restricted contrast is continued. Otherwise, output the image directly. The mathematical expression of bilateral filtering is as follows:

$$I'(x) = \frac{1}{W_p} \sum_{x_i \in \Omega} I(x_i) f(\|x_i - x\|) g(\|I(x_i) - I(x)\|)$$
(3)

Where, $f(\cdot)$ is the spatial distance weight function, $g(\cdot)$ is the pixel intensity weight function, $W_p$ is the normalization coefficient.

Contrast calculation formula is as follows:

$$T = \frac{\sigma_I}{\mu_I}$$
(4)

Where, $\sigma_I$ and $\mu_I$ are the standard deviation and mean of the image respectively to quantify the basis for contrast adjustment.

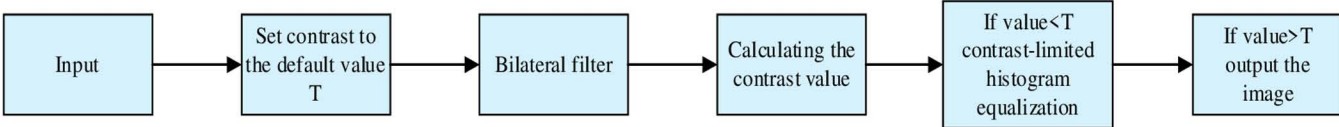

**Fig 2. Filter equalization enhancement algorithm flow.**

The distance function and kernel function are two important functions that constitute the bilateral filter. The weighted average value of the target pixel and the surrounding pixel is calculated to achieve the filter, which can effectively smooth the image and remove the noise while preserving the sharpness of the image edge. The histogram equalization is realized by adjusting the gray distribution of the image to improve the contrast, and the constraint parameters are added to reduce the amplification of noise and the loss of detail. Finally, the enhanced crack image generated by this algorithm has higher visual clarity and better crack feature expression, which is suitable for real-time crack identification and detection requirements of intelligent seam filling machines.

**2.2.3. Data enhancement processing comparison.** In image recognition research, the quality of data set directly affects the performance of the model, so it is necessary to pre-process the crack data set effectively before crack recognition. As a common image processing method [19], traditional histogram equalization can improve the contrast and visual effect of images, but its performance in pavement crack detection is relatively general, and it is difficult to highlight the detailed characteristics of cracks. Therefore, an image enhancement technology is proposed in this study, and its effect comparison is shown in Fig 3.

Fig 3 shows the comparison of the pre-processing effects of irregular road cracks in the self-made data set. The original block crack image, the image after histogram equalization and the image of mean filtering operation are listed respectively. The traditional histogram equalization method equalizes the whole image globally, but does not fully consider the details of the image, which leads to the obvious increase of noise and the crack information is covered by other features.

In order to overcome the shortcomings of traditional histogram equalization methods, a filter equalization enhancement algorithm is proposed in this paper. Firstly, the crack image is filtered bilaterally to effectively remove noise while preserving edge details. Then histogram equalization combined with contrast restriction is applied to precisely control the enhancement of image contrast. Noise amplification and excessive enhancement have always been the problems of traditional methods, so the limitation of contrast parameters can be avoided, which significantly improves the image processing effect and the crack recognition ability of the model.

In this experiment, a comparative experiment method was adopted. The unimproved YOLOv8 model was used to train on the original data set and the equalized and filtered data set respectively. Other parameters were consistent, and the accuracy rate obtained from the two trainings was compared. The training results before and after equalization filtering are shown in Table 2.

It is not difficult to see from Table 2 that the recognition effect achieved by using equalization filtering to enhance the data set is better than that achieved by the original data set.

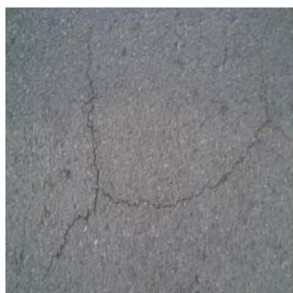   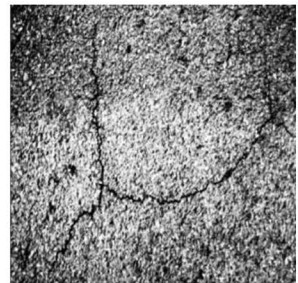   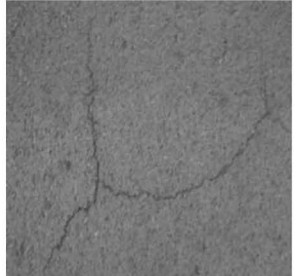

(a)Initial image          (b) Histogram equalization          (c) Mean filtering

**Fig 3. Comparison of pretreatment effect.**

## 3. Detection algorithm improvement

### 3.1. Algorithm overview and detection process

YOLOv8 is a classic version of the YOLO family of algorithms, It combines more lightweight backbone networks such as CSPDarknet and the effective multi-scale Feature fusion method PANet (Path Aggregation Network) and FPN (Feature Pyramid) in neck Network) and anchor-free detection strategy of head part, which has strong real-time and detection ability. Through the collaborative work of backbone, neck and head [20], YOLOv8 has found a good balance between accuracy and speed, and has a wide range of application prospects in the field of target detection.

YOLOv8 algorithm was used to train the self-made data set. The model training process in this study is shown in Fig 4. First, the data in the open data set rdd2022 was selected and combined with the data collected by the author on site. labelimg annotation tool [21] was used to label the pavement crack image and produce the data set that could be used for yolo training. Secondly, the data set is divided into training set and test set, and the training set is processed by equalization filtering. Finally, the model is trained, and the model is tested on the test set, so as to obtain the crack recognition results of the model.

### 3.2. YOLOv8 model improvements

The improved model YOLOV8-DGS proposed in this study is optimized for the computational complexity and feature extraction capability of the YOLOv8 model, and focuses on improving the backbone and neck structures. On the premise of keeping the number of layers unchanged, we introduce Depthwise Separable Convolution (DWConv) and Ghost Shuffle Convolution (GSConv). To reduce the calculation and improve the detection accuracy.

YOLOv8's backbone is mainly used to extract feature information of different scales. Standard Convolution takes a lot of computation and has high redundancy in feature extraction. In order to improve computational efficiency, we replace the standard convolution modules at layers 0, 1, 3, and 5 in backbone with DWConv (light blue fills the modular part, as

**Table 2. The training results before and after filtering equalization.**

| Fracture type | Equalizing filter before (%) | After equalizing filter (%) |
|---|---|---|
| Longitudinal | 61.5 | 74.7 |
| Transverse | 60.0 | 72.5 |
| Block | 68.5 | 78.6 |
| Irregularity | 62.5 | 70.9 |

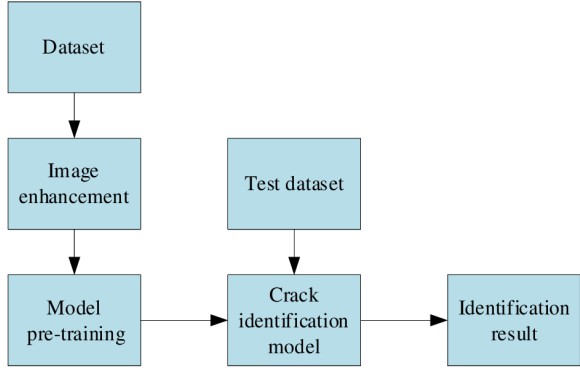

**Fig 4. Pavement crack recognition process based on YOLOv8 modeling.**

shown in Fig 5). The core idea of DWConv is to split standard Convolution into Depthwise Convolution and Pointwise Convolution. Channel by channel convolution is only carried out in a single channel, rather than across channels like standard convolution, so as to reduce the computation. Point-by-point convolution is the use of 1×1 convolution to aggregate information across channels to restore feature representation capabilities. With the introduction of DWConv, backbone is able to reduce parameters and computation while maintaining strong feature extraction capabilities, thus processing crack texture information more efficiently.

YOLOv8's neck is mainly used for multi-scale feature fusion, usually using upsampling and downsampling operations to enhance the robustness of target detection. Traditional convolution is computationally complex in this part, so we replace all standard convolution with GSConv in the lower sample (orange fill module, as shown in Fig 5). GSConv divides input channels into multiple groups by Grouped Convolution, and convolution operations are carried out independently within each group, thus reducing computational complexity. Shuffle Operation The Shuffle operation uses the Channel Shuffle mechanism to enable the features of different groups to interact with each other to compensate for information loss caused by group convolution. The introduction of GSConv enables neck to fuse information at different scales more efficiently, while reducing computational costs, improving the effectiveness of feature transmission, and enhancing the accuracy of crack detection.

**3.2.1. DWConv depth-wise convolution.** In the traditional convolution operation, it is assumed that the size of the input feature graph is $H \times W \times Cin$, and the size of the output feature graph is $H \times W \times Cin$, where $Cin$ is the number of

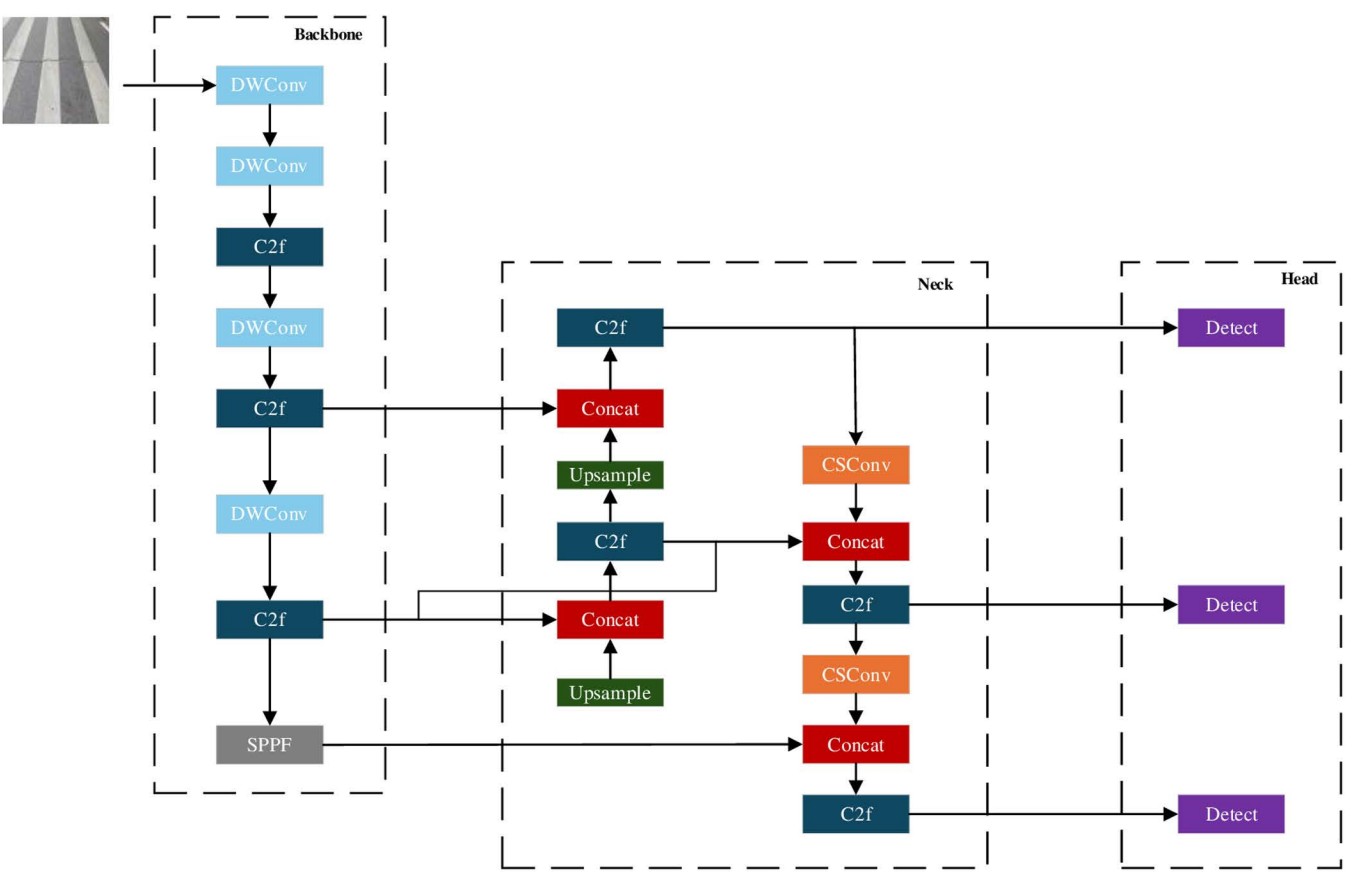

**Fig 5. Structure of YOLOv8-DGS.**

input channels and *Cout* is the number of output channels. For each input channel and output channel, a convolution calculation is performed. Assuming the size of the convolution kernel is $K \times K$, then the number of parameters for each convolution layer will be:

$$\text{Number of parameters} = K \times K \times Cin \times Cout$$

As the number of input and output channels increases, the amount of computation and the number of parameters increases dramatically, which makes deep neural networks very time consuming when training and reasoning, especially on mobile or embedded devices.

In this case, it is necessary to introduce Depthwise Convolution [22], which significantly reduces computational complexity by decomposes traditional Convolution operations into two simpler operations: Depthwise convolution and Pointwise Convolution (1x1 convolution).

Deep convolution operates on each input channel separately, rather than across channels as with traditional convolution. That is, for each channel of the input, an independent convolution kernel is used for convolution calculation, and the convolution operations of different channels do not interfere with each other. The characteristic of this operation is that the size of the convolution kernel is still $K \times K$, but each convolution kernel is only responsible for the corresponding input channel, and is no longer combined with other channels as is the case with standard convolution. The amount of computation is greatly reduced, and the number of parameters is also greatly reduced. After deep convolution, point-by-point convolution (i.e., 1×1 convolution) is usually used to fuse the individual input channels into the output channels. This step is responsible for linear combination of each output channel after deep convolution to generate the final output channel. The function of point-by-point convolution is to combine the features of multiple channels generated by the deep convolution operation to restore the channel dimension of the feature graph.

For a deeply separable product, we first perform a deep convolution (convolution for each input channel):

$$\text{Number of depth convolution parameters} = K \times K \times Cin$$

Then, use point-by-point convolution to merge features:

$$\text{Number of point by point convolution parameters} = Cin \times Cout$$

Thus, the total number of parameters for depth separable integrations is:

$$\text{Depth separable volume total parameter number} = K \times K \times Cin + Cin \times Cout$$

In this way, the depth divisible volume greatly reduces the number of parameters and the amount of computation in the model. The depth-separable convolution module and its operation process are shown in Fig 6.

Unlike traditional convolution, depth-separable convolution changes its internal computational structure while keeping the dimensions of the input and output images unchanged. Therefore, depth-separable convolution can perfectly replace traditional convolution without adding or reducing the number of backbone layers. Specifically, deep-separable convolution enhances the range of receptive fields by decomgenerating convolution operations so that the convolution kernel of each channel only processes data from a single channel [23]. This method helps the model to capture the detail information in the crack image better, so the accuracy and efficiency of the target detection are greatly improved.

**3.2.2. GSConv group shuffle convolution.** GSConv (Group-shuffle Conv) [24] (Fig 7) is an improved convolution operation, which aims to solve the problem of excessive computational load and parameters of convolution operations in deep neural networks, while improving the computational efficiency and expressiveness of the model. It combines Group

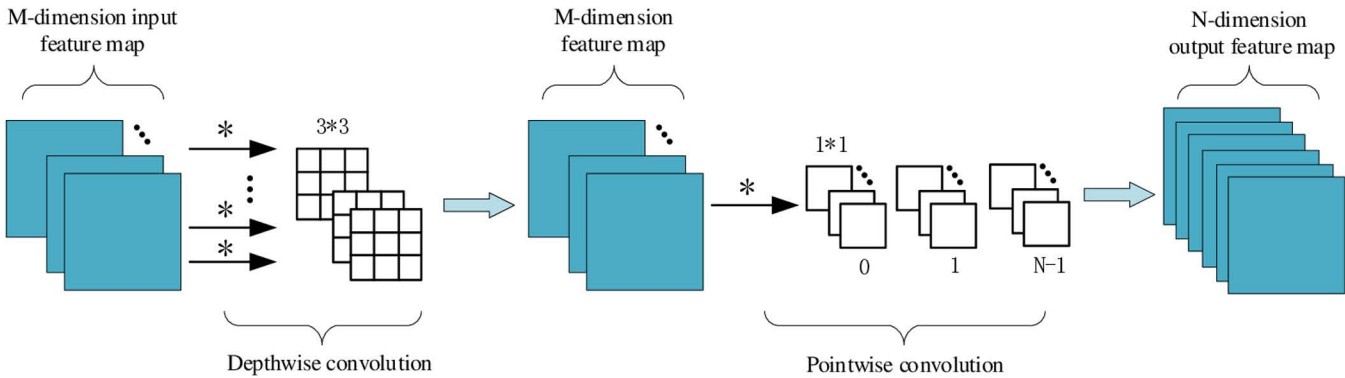

**Fig 6. Structure diagram of depth-separable volume integration algorithm.**

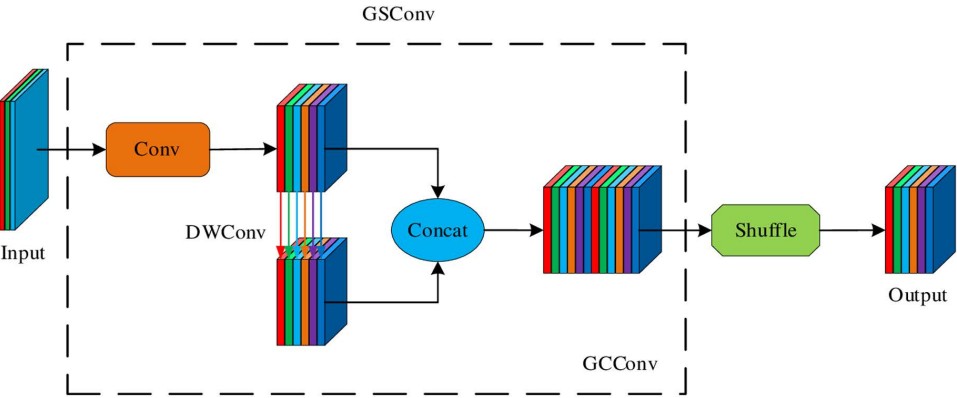

**Fig 7. GSConv (Group-shuffle Conv) structure diagram.**

Convolution and Channel Shuffle techniques to enhance the network's learning ability while keeping the computational effort low, especially when dealing with tasks such as small object detection.

In traditional convolution operation, the input feature map and the convolution kernel are convolved to generate the output feature map. As the depth of the network and the dimension of the input feature graph increase, the computation and parameter number of the convolution operation also increase exponentially. In order to reduce the computation and parameter number, grouping convolution comes into being. The core idea of grouping convolution [25] is to divide the channels of input feature graphs into multiple groups, and each group performs convolution operations separately to reduce the amount of computation. However, although grouping convolution can effectively reduce the computation and parameter number, it also has the limitation that there is no information exchange between the channels in each group, resulting in the loss of feature expression ability. In order to overcome this problem, channel rearrangement technology comes into being. By rearranging the channels in each group in a specific order, GSConv ensures efficient information exchange between each channel. This enables the model to obtain richer feature representations while maintaining low computational complexity.

GSConv (Group-shuffle Conv) introduces the channel rearrangement technology, which disrupts and rearranges the connections between channels after a grouping convolution operation, thus facilitating the flow and sharing of information across groups. Specifically, GSConv consists of the following two key steps:

Group convolution: First, the input feature graph is divided into multiple groups, and each group is independently con-volution operation, which can reduce the calculation amount and the number of parameters of each group of convolution operation.

Channel Shuffle: After the packet convolution is complete, the channel is rearranged. That is, the channels in each group are rearranged according to certain policies, so that the channels from different groups can be connected to each other, so as to realize the information flow across groups. This helps enhance feature representation.

The improvement of feature extraction ability of the improved model compared with the original model is shown in Fig 8. The FLOPs, number of parameters and number of layers of the improved model and the comparison experimental group model are shown in Table 3.

## 4. Experiment

### 4.1. Experimental parameter settings

The hardware parameters of this study are a desktop computer provided by the working unit, the central processor is the Core i5-12400f launched by Intel Corporation, the GPU used for graphics processing is the RTX4070 launched by NVIDIA, and the running memory is 16GB, which can better meet the model training work. Fig 9 shows the model training

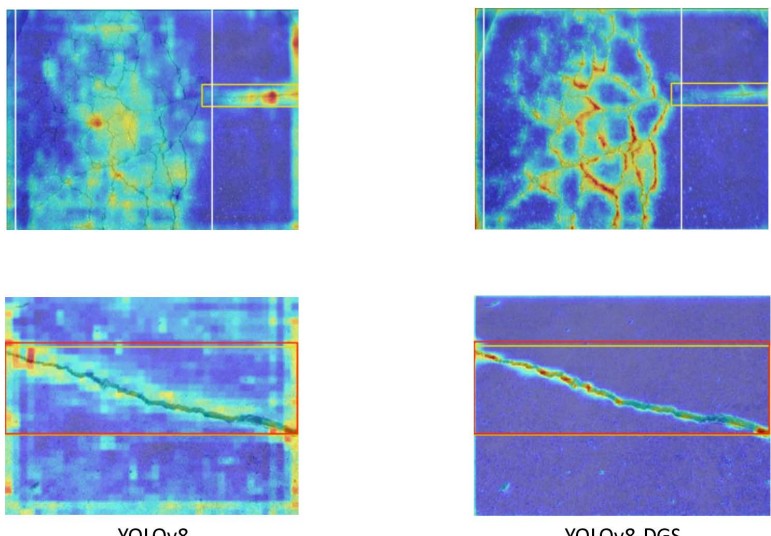

YOLOv8              YOLOv8-DGS

**Fig 8. Comparison of feature extraction heat map before and after the improved algorithm.**

**Table 3. Different model parameters and test results.**

| Model | FLOPs (B) | Params (M) | Layers |
|---|---|---|---|
| YOLOv5s | 15.9 | 7.21 | 213 |
| YOLOv8n | 8.7 | 3.20 | 168 |
| YOLOv8s | 28.6 | 11.2 | 168 |
| YOLOv8-DW | 7.3 | 10.13 | 187 |
| YOLOv8-GS-slim | 23.0 | 8.88 | 266 |
| YOLO11s | 21.7 | 9.45 | 319 |
| YOLOv8-DGS | 24.3 | 9.20 | 174 |

samples, which are composed of multiple pictures of the same size. Before training, these pictures will undergo data-enhanced operations with the model itself, such as simple rotation, scaling and cropping operations to increase the generalization ability and stability of the model.

The experiment used annaconda to create a virtual environment running on the pycharm compilation platform, python version 3.10. Table 2 lists the parameter values of this experiment. The number of training rounds is set to 200, which can ensure the stability of the training results. Because the GPU with good performance is used, the batch is set to 32. The input size is set to 640 consistent with the data set size, and the Mosaic is set to 0 because mean filtering was used to enhance the data set in this study. In the training process, the number of model layers, the number of parameters, the calculation complexity, mAP50 and MAP50-95 are calculated and output to show the performance of the model in the training process. Table 4 shows the specific parameters.

## 4.2. Evaluation index

In order to comprehensively evaluate the performance of the improved YOLOv8 model in pavement crack detection, this study adopted a variety of common target detection and evaluation indexes. These indicators can objectively reflect the performance of the model in different aspects, including detection accuracy, recall rate, positioning ability and speed. Specific evaluation indicators include Precision, Recall, F1-score, mAP and Inference Time, etc. [26].

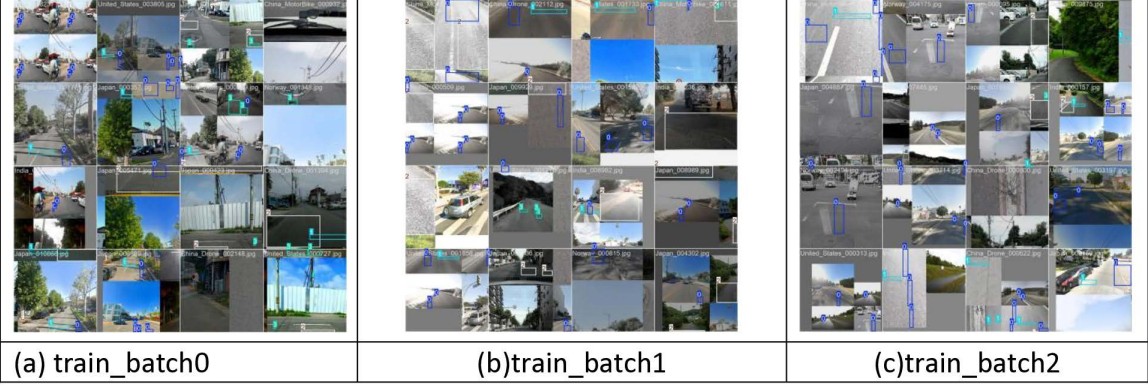

| (a) train_batch0 | (b)train_batch1 | (c)train_batch2 |

**Fig 9. Training visualization.**

**Table 4. Hyper-parameter Settings.**

| Parameter | Settings |
| --- | --- |
| Initial learning rate(lr0) | 0.01 |
| Final learning rate(lrf) | 0.02 |
| Batch size | 32 |
| Weight_decay | 0.001 |
| Warmuo_epochs | 3.0 |
| Warmup_bias_lr | 0.1 |
| Epochs | 100 |
| Mosaic | 0 |

Accuracy is the proportion of positive samples predicted correctly in the model's detection results, and recall measures the ability of the model to identify all actual cracks, i.e., the proportion of true positives to all actual cracks. The formula is as follows:

$$P = \frac{TP}{TP+FP} \tag{5}$$

$$R = \frac{TP}{TP+FN} \tag{6}$$

Where $TP$ is true positive (number of crack areas correctly detected), $FP$ is false positive (number of crack areas incorrectly detected), and $FN$ is false negative (number of crack areas missed). The high accuracy means that most of the crack areas detected by the model are correct, that is, the false positive rate is low. The high recall rate means that the model detects as many cracks as possible, reducing missed detection.

F1-score is the harmonic average of accuracy and recall rate, which is an important index to measure the comprehensive performance of the model. By considering both accuracy and recall, it avoids the bias of considering one metric in isolation. Its calculation formula is as follows:

$$F1 - \text{score} = 2 \times \frac{P \times R}{P+R} \tag{7}$$

F1-score can balance accuracy and recall rate to a certain extent, and it is a comprehensive index to evaluate the performance of detection models.

mean Average Precision (mAP) is one of the most commonly used evaluation indexes in target detection, especially in multi-class detection tasks. mAP comprehensively reflects the detection ability of the model by calculating the average accuracy under different recall rates. Specifically, mAP gets the final result by calculating the average accuracy (AP) for each category and then averaging the AP across all categories. The formula is as follows:

$$mAP = \frac{1}{N} \sum_{i=1}^{N} APi \tag{8}$$

Where N is the total number of classes, and $APi$ is the average precision of the I-th class. The higher the mAP value, the better the overall detection performance of the model.

In addition to the traditional indexes such as precision and recall rate, the computational efficiency in the target detection task is also an important evaluation criterion. Detection speed is usually measured by Inference Time, which is the time it takes a model to process an image. The shorter the reasoning time, the better the performance of the representation model in real-time applications, the faster the crack detection can be performed, and the real-time detection scenarios such as intelligent sewing machines can be adapted.

## 4.3. Experimental results

The control group of this implementation is ultralytics company's classic YOLO model. YOLOv5s and YOLOv8n, which have a small number of parameters, are selected. Due to the reduction of the number of parameters, the reasoning speed of the model is improved and the accuracy is still good, but it is not enough to be applied to intelligent engineering equipment. YOLO11, launched this year by ultralytics, also shows good potential, achieving 65.2% accuracy on this dataset, but the downside of the improved accuracy is that the running speed is reduced, and the large number of parameters and computation are not suitable for practical devices. The reason for the improvement on the basis of YOLOv8s is that it takes into account the speed and accuracy of recognition. By integrating the improved methods of YOLOv8 proposed

at the present stage, such as depth-separable convolutional module and spatial depth conversion convolutional module, traditional convolutional modules in Backbone network and Neck are replaced, and comparative experiments are set. Although these modules show better recognition accuracy on self-made data sets after replacement, the inference speed is still not satisfactory. The best result is that GSConv and DWConv are embedded in the network, and the YOLOv8-DGS network is obtained on the experimental data set. Table 5 shows the comparison of experimental results between the improved model and other models. The mAP50 and MAP50-95 of YOLOv8-DGS reached 91.6% and 61.3% respectively, which increased by 13.5% and 15.2% compared with the YOLOv8s model. Meanwhile, the parameters of the improved model were reduced by 11.6% compared with the original model, which improved the detection speed. Fig 10 shows the identification results of four different road cracks.

Table 5. Different model test results.

| Model | mAP50 | mAP50-95 |
|---|---|---|
| YOLOv5s | 0.624 | 0.323 |
| YOLOv8n | 0.604 | 0.412 |
| YOLOv8s | 0.644 | 0.428 |
| YOLOv8-DW | 0.841 | 0.506 |
| YOLOv8-GS-slim | 0.812 | 0.476 |
| YOLO11s | 0.626 | 0.431 |
| YOLOv8-DGS | 0.916 | 0.613 |

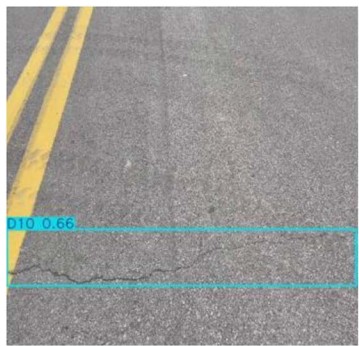
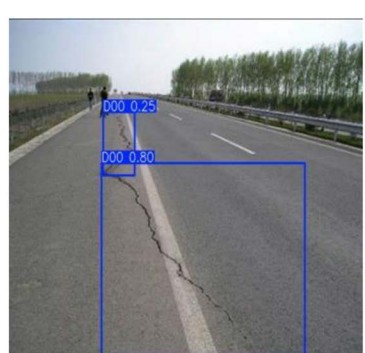
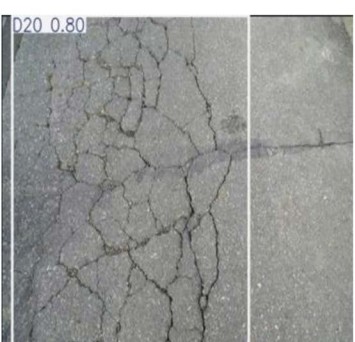
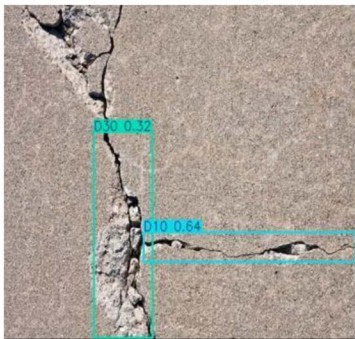

Fig 10. Test results of YOLOv8-DGS model.

In order to prove the superiority of the improved model, statistical analysis, especially P-value calculation, is needed to measure the significance of the experimental results. Detection indicators follow normal distribution, independent sample t test can be used:

$$t = \frac{\bar{X}_1 - \bar{X}_2}{\sqrt{\frac{s_1^2}{n_1} + \frac{s_2^2}{n_2}}}$$

(9)

Where, $\bar{X}_1$, $\bar{X}_2$ is the average performance of YOLOV8-DGS and standard YOLOv8 respectively, $s_1$, $s_2$ is the standard deviation, and $n_1$, $n_2$ is the sample number. Table 6 shows the T-test P-values of the improved model and the original model under the indexes mAP50 and MAP50-95, of which both P-values are less than 0.05, indicating that the improvement in detection performance of the improved model YOLOv8-DGS is not accidental, but statistically significant, and it can be considered that the improved model is superior to YOLOv8s in reliability.

Faster running speed is an important condition for application in smart devices [27]. In this experiment, FPS was used to characterize the inference speed of the model. The formula for calculating FPS is shown in formula (10).

$$FPS = \frac{1000}{pre + inference + post}$$

(10)

Among them, pre is the pre-processing time of the model, which is to convert the data into a format suitable for training to reduce the computing load and help improve the performance of the model; inference is the inference time of the model, that is, the time used to pass the preprocessed data into the model and output it; post is the post-processing time of the model, which is the time spent decoding the output and converting the format. The units of the three are milliseconds. FPS is the time it takes to run all the above processes.

Table 7 shows the inference speed comparison of several typical models. These models all use smaller models such as YOLOv5s and YOLOv8n with fewer parameters, and the control group of other models is also a lightweight improved model. Compared with these models, the inference speed of YOLOv8-DGS in this study is the first in the case of approximate number of parameters, reaching 85 FPS. It is directly proved that the improved model can improve the reasoning speed on the premise of ensuring the recognition accuracy, and provide the feasibility for the subsequent deployment to the intelligent sewing machine.

Table 6. P-value analysis of YOLOV8-DGS compared with YOLOv8.

| Index | t tests the p-value |
|---|---|
| mAP50 | $1.03 \times 10^{-7}$ |
| mAP50-95 | $8.46 \times 10^{-7}$ |

Table 7. Comparison of the running speed of various models.

| Model | Pre-process(ms) | Inference(ms) | Post-process(ms) | FPS |
|---|---|---|---|---|
| YOLOv5s | 0.5 | 11 | 1.4 | 74 |
| YOLOv8n | 0.5 | 9.7 | 1.7 | 83 |
| YOLOv8s | 0.7 | 10.3 | 1.8 | 78 |
| YOLOv8-DW | 0.3 | 10.5 | 1.6 | 80 |
| YOLOv8-GS-slim | 0.4 | 10.3 | 1.4 | 84 |
| YOLO11s | 0.6 | 10.1 | 1.7 | 80 |
| YOLOv8-DGS | 0.5 | 10.1 | 1.6 | 85 |

**Table 8.  Ablation test results.**

| Backbone (DWConv replaces Conv) | | | | Neck (GSConv replaces Conv) | Evaluation index | | |
|---|---|---|---|---|---|---|---|
| 0 | 1 | 3 | 5 | All | Precision | Recall | mAP |
| − | − | − | − | − | 0.722 | 0.74 | 0.71 |
| − | − | − | − | √ | 0.768 | 0.70 | 0.75 |
| − | − | − | √ | √ | 0.812 | 0.79 | 0.84 |
| − | − | √ | √ | √ | 0.844 | 0.81 | 0.86 |
| − | √ | √ | √ | √ | 0.902 | 0.89 | 0.84 |
| √ | √ | √ | √ | √ | 0.916 | 0.90 | 0.92 |

## 4.4.  Ablation experiment

The ablation experiment aims to verify whether the YOLOv8-DGS model has improved the detection efficiency of crack-like data set compared with the original model. The specific experimental method is the control variable method, which replaces the convolution in Neck on the basis of the original model, and compares the recognition accuracy of traditional convolution and grouping shuffle convolution with the same backbone. After the recognition effect is improved, the Neck part is kept unchanged, and the convolution of layers 0, 1, 3 and 5 in backbone are replaced successively, and the recognition accuracy of depth-separable convolution and traditional convolution are compared. Other parameters remain the same. The experimental results are shown in Table 8.

As can be seen from Table 8, the conventional convolution of layers 0, 1, 3 and 5 of the YOLOv8 model is replaced by deep separable volume, while the convolution modules in the Neck structure are all replaced by grouping shuffle convolution. The average accuracy of the above method is the highest, reaching 91.6%. It can be seen that the improved YOLOV8-DGS model based on the YOLOv8 model has better performance.

## 5.  Conclusion

This paper proposes a road crack detection algorithm based on improved YOLOv8, which is used to detect complex pavement and different types of pavement defects that are difficult to distinguish due to obstacles. This method makes good preparation for subsequent work such as defect location, slot opening and joint filling. The algorithm improves two parts: First, deep separable convolution is introduced into the third, fifth and seventh layers of the backbone network, and the number of parameters is reduced through the operation of separated convolution, and the ability of crack feature learning is enhanced. On the other hand, by replacing the traditional convolution in the Neck part of yolov8, the calculation amount of the model is greatly reduced, the detection accuracy of the model is improved, and the detection speed is accelerated. In order to prove the effectiveness of the improved model, the author conducted validation on 19512 data sets. Through model training and verification, the effectiveness of the proposed method is confirmed.

Although the improved model has achieved a good detection effect, the original image quality of the data set used for training is not high, and the types of pavement defects are not covered, so the generalization ability of the model still needs to be improved. There are also challenges in identifying small cracks and pothole-type damage on shaded or wet roads.

In the follow-up work, the research will continue to collect higher-quality pavement crack images to train the improved model and further improve the generalization ability of the model. In order to better apply the model to the intelligent filling equipment, the subsequent research focus is on the lightweight of the model. Pruning techniques such as Max-Min and alpha-beta are used to compress the model, and accelerated deployment techniques such as TensorRT are combined to apply the model to embedded devices.

## Acknowledgments

The authors sincerely thank the Editor-in-Chief, and the anonymous reviewers for their detailed comments and constructive suggestions, which greatly improved this manuscript.

## Author contributions

**Data curation:** ZuXuan Zhang.

**Funding acquisition:** HongLi Zhang, TongJia Zhang.

**Software:** ZuXuan Zhang.

**Supervision:** HongLi Zhang, TongJia Zhang.

**Writing – original draft:** ZuXuan Zhang, HongLi Zhang.

**Writing – review & editing:** ZuXuan Zhang, HongLi Zhang.

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
