## [Decision Letter · Decision Letter 0]

25 Mar 2025

PONE-D-25-10806A pavement crack identification method based on improved yolov8PLOS ONE

Dear Dr. Zhang,

Thank you for submitting your manuscript to PLOS ONE. After careful consideration, we feel that it has merit but does not fully meet PLOS ONE’s publication criteria as it currently stands. Therefore, we invite you to submit a revised version of the manuscript that addresses the points raised during the review process.

We look forward to receiving your revised manuscript.

Kind regards,

Ahmed M. Yosri

Academic Editor

PLOS ONE

“Natural Science Foundation project of Shandong Province

Project number: ZR2024QE374

Project name: Research on the key technology of robot six-degree-of-freedom grasping and detecting of highly reflective parts in unordered stacking scene.”

“This research was supported by the Natural Science Foundation project of Shandong Province (ZR2024QE374).”

“Natural Science Foundation project of Shandong Province

Project number: ZR2024QE374.”

Reviewers' comments:

Reviewer's Responses to Questions

**Comments to the Author**

1. Is the manuscript technically sound, and do the data support the conclusions?

Reviewer #1: Partly

Reviewer #2: Yes

Reviewer #3: Yes

2. Has the statistical analysis been performed appropriately and rigorously? 

Reviewer #1: No

Reviewer #2: No

Reviewer #3: Yes

3. Have the authors made all data underlying the findings in their manuscript fully available?

Reviewer #1: Yes

Reviewer #2: Yes

Reviewer #3: Yes

4. Is the manuscript presented in an intelligible fashion and written in standard English?

Reviewer #1: Yes

Reviewer #2: Yes

Reviewer #3: Yes

5. Review Comments to the Author

Reviewer #1: The manuscript deals with " A pavement crack identification method based on improved yolov8." The following comments and suggestions need to be addressed before its consideration.

1. The manuscript should provide a more detailed explanation of the YOLOv8-DGS model improvements. Specifically, elaborating on how deep separable convolution (DWConv) and GSConv are integrated into the backbone and neck of the model would help readers understand the technical advancements made in crack detection

2. These references could be cited in the introduction, particularly when discussing the advancements in deep learning methods for image processing and detection tasks. It would fit well in a paragraph that highlights the growing application of deep learning in various fields, including remote sensing and crack detection. Remote Sensing, 15(10), 2663. https://doi.org/10.3390/rs15102663, Ocean Engineering, 259, 111735. https://doi.org/10.1016/j.oceaneng.2022.111735

3. This reference can be cited in the introduction, particularly when discussing advancements in the YOLOv8 model, machine learning, and its applications in various fields, including marine debris detection. It would be relevant to mention this study when highlighting the versatility and improvements of YOLOv8 in different contexts. Journal of Marine Science and Engineering, 12(10), 1748. https://doi.org/10.3390/jmse12101748, International Journal of Computational Methods, 2450066. 10.1142/S021987622450066X, Journal of Computing in Civil Engineering, 39(3), 4025017. 10.1061/JCCEE5.CPENG-6167, Scientific Reports, 14(1), 15170. 10.1038/s41598-024-66234-3, Measurement Science and Technology, 36(1), 015104. 10.1088/1361-6501/ad7f77, GPS Solutions, 28(4), 178. 10.1007/s10291-024-01715-6, IEEE Transactions on Vehicular Technology, 1-16. 10.1109/TVT.2024.3492388

4. It is essential to include a comprehensive description of the experimental setup used to evaluate the YOLOv8-DGS model. This should encompass the dataset characteristics, the sample number, and the test case selection criteria. Providing this information will enhance the reproducibility of the study

5. While the manuscript mentions impressive performance metrics such as Precision, Recall, F1-score, and mAP50, it would be beneficial to include a comparison with existing models, such as YOLOv3 or other state-of-the-art methods. This comparison can highlight the advantages of the proposed model and provide context for the reported results

6. These references could be cited when discussing the importance of real-time identification systems in various environments, including underground disaster areas. It would be relevant to mention this study while highlighting the advancements in detection technologies and their applications in safety and rescue operations. A potential sentence could be: "Recent studies have demonstrated the effectiveness of multi-source heterogeneous data fusion in real-time identification of critical situations in disaster areas, which aligns with the advancements in detection technologies discussed in this paper. Safety Science, 181, 106690. https://doi.org/10.1016/j.ssci.2024.106690. Remote Sensing, 15(17), 4251. https://doi.org/10.3390/rs15174251. Advances in Civil Engineering, 2023(1), 8897139. https://doi.org/10.1155/2023/8897139. Chinese Journal of Mechanical Engineering, 37(1), 108. 10.1186/s10033-024-01107-4. Computers and Geotechnics, 178, 106949. https://doi.org/10.1016/j.compgeo.2024.106949. Computers and Geotechnics, 177, 106827. https://doi.org/10.1016/j.compgeo.2024.106827

7. Incorporating visual aids, such as graphs or tables, to present the performance metrics and comparisons with other models can significantly improve the manuscript's clarity and impact. Visual representations can help readers quickly grasp the improvements made by the YOLOv8-DGS model

8. The manuscript should address any limitations of the proposed method. Discussing potential challenges, such as the model's performance in varying environmental conditions or with different types of pavement materials, would provide a balanced view of the research

9. Including a section on future work could enhance the manuscript. Suggestions for further research, such as exploring the application of the YOLOv8-DGS model in real-time scenarios or integrating it with other technologies, would demonstrate the ongoing relevance of the research

10. A thorough manuscript proofreading is recommended to ensure clarity and coherence. This includes checking for grammatical errors, ensuring consistent terminology, and improving the overall flow of the text

Reviewer #2: I have reviewed the manuscript entitled " A pavement crack identification method based on improved yolov8" the manuscript is insightful and is structured well, there are several areas that require further refinement. Below, I have provided section-wise review with specific comments to enhance the clarity, rigor, and overall impact of your work.

The title is clear and informative but could be slightly refined. Suggested improvement:

"Enhanced YOLOv8-Based Pavement Crack Detection: A High-Precision Approach".

Consider explicitly mentioning "Deep Learning" or "Computer Vision" to attract a broader audience.

In the abstract section add a sentence highlighting how this method outperforms conventional YOLO models. The section lacks a direct comparison with existing YOLO models. It also does not explicitly mention real-time feasibility or computational efficiency. The claim about "improving performance" is general—what aspect? Accuracy, speed, or robustness?

Specify if this model is suitable for real-time deployment.

In the introduction section, the paper mentions manual inspection is inefficient but does not provide statistics or real-world implications (e.g., cost/time of manual inspection vs. AI-based detection). The problem statement could be more focused. Instead of stating that "YOLOv8 has potential," explicitly highlight its limitations in pavement crack detection. The introduction does not clearly state YOLOv8 alone is insufficient. Instead of saying "YOLOv8 has potential," it should specify its weaknesses (e.g., handling small cracks, computational load). Other crack detection methods (e.g., transformer-based approaches) are missing. How does this work compare to recent works like CrackFormer? The paper should explicitly state the gap in previous research that this study addresses.

In the material and method section, the sample size and its justification need more emphasis. A dataset split ratio (80:20) is mentioned, but no rationale is provided. Does the dataset cover diverse road conditions (e.g., lighting, shadows, different pavement materials)? What resolution are the images? Are there any preprocessing steps to remove blur/noise? Will a model trained on RDD2022 generalize well to different environments?

The section could benefit from a statistical analysis of how preprocessing improved accuracy.

Section 2.2… How does preprocessing affect accuracy? A small table comparing performance before/after preprocessing would strengthen this section. Mean filtering is explained but could use more mathematical rigor. Why was mean filtering chosen? Does it outperform Gaussian filtering or median filtering for noise reduction?

Section 3… Detection algorithm improvement… The paper claims the improvements reduce computational load, but where is the evidence? A table comparing FLOPs and parameter count before and after modification is needed. How do the modifications affect feature extraction? A grad-CAM visualization would strengthen this section.

In the experimental section of the paper, were hyperparameters optimized via grid search? Why were specific values chosen? Would adding more layers improve accuracy?

Section 4.2…. Are differences statistically significant? A p-value analysis is needed. A statistical test comparing YOLOv8-DGS with standard YOLO models would strengthen claims.

"Inference Time" should be broken down into GPU vs CPU performance.

Section 5… Conclusion section lacks a discussion of failure cases: When does the model fail? Low contrast? Blurry images?

Future work is vague: What specific lightweight techniques will be explored?

No discussion of practical deployment: How can this be integrated into real-world applications?

Reviewer #3: This paper discusses crack detection, which is very important for road management. An image enhancement technology and a recognition algorithm are developed to enhance the performance of YOLO 8. The developed method YOLOv8-DGS can effectively identify the cracks on pavement.

(1) How do you measure the “accuracy” in the case study? It is the most important criterion. Its result is given without calculation or definition. How do you know the actual measurement of the cracks?

(2) Compared to YOLO 11, YOLOv8-DGS has better accuracy on mAP50 but worse accuracy on MAP50-95. Their running speeds are also similar. Does it mean YOLOv8-DGS is not better than YOLO 11? Please explain.

(3) This paper has many typos and errors, for example, it should be Table 3 instead of Figure 3 on Line 354. Please carefully revise this paper before publication.

6. PLOS authors have the option to publish the peer review history of their article (what does this mean? ). If published, this will include your full peer review and any attached files.

**Do you want your identity to be public for this peer review?** For information about this choice, including consent withdrawal, please see our Privacy Policy .

Reviewer #1: No

Reviewer #2: No

Reviewer #3: No

---

## [Author Response · Author response to Decision Letter 0]

6 Apr 2025

We have carefully read the comments of reviewers and editors, and made changes and replies in response to these comments. The information for the Response has been submitted under "Response to Reviewers".

---

## [Decision Letter · Decision Letter 1]

28 Apr 2025

Enhanced YOLOv8-Based Pavement Crack Detection: A High-Precision Approach

PONE-D-25-10806R1

Dear Dr. Zhang,

We’re pleased to inform you that your manuscript has been judged scientifically suitable for publication and will be formally accepted for publication once it meets all outstanding technical requirements.

Kind regards,

Ahmed M. Yosri

Academic Editor

PLOS ONE

Additional Editor Comments (optional):

Reviewers' comments:

Reviewer's Responses to Questions

**Comments to the Author**

1. If the authors have adequately addressed your comments raised in a previous round of review and you feel that this manuscript is now acceptable for publication, you may indicate that here to bypass the “Comments to the Author” section, enter your conflict of interest statement in the “Confidential to Editor” section, and submit your "Accept" recommendation.

Reviewer #1: All comments have been addressed

Reviewer #2: All comments have been addressed

2. Is the manuscript technically sound, and do the data support the conclusions?

Reviewer #1: Yes

Reviewer #2: Yes

3. Has the statistical analysis been performed appropriately and rigorously? 

Reviewer #1: Yes

Reviewer #2: N/A

4. Have the authors made all data underlying the findings in their manuscript fully available?

Reviewer #1: Yes

Reviewer #2: Yes

5. Is the manuscript presented in an intelligible fashion and written in standard English?

Reviewer #1: Yes

Reviewer #2: Yes

6. Review Comments to the Author

Reviewer #1: The content of paper was well organized, all the suggested points are incorporated and easy for the reader to follow the subject discussed, thus support for its acceptance.

Reviewer #2: I am satisfied with responses, the authors have incorporate all the changes and addressed all the points, so the paper can be accepted for publication.

7. PLOS authors have the option to publish the peer review history of their article (what does this mean? ). If published, this will include your full peer review and any attached files.

**Do you want your identity to be public for this peer review?** For information about this choice, including consent withdrawal, please see our Privacy Policy .

Reviewer #1: No

Reviewer #2: No

---

## [Editor Report · Acceptance letter]

PONE-D-25-10806R1

PLOS ONE

Dear Dr. Zhang,

I'm pleased to inform you that your manuscript has been deemed suitable for publication in PLOS ONE. Congratulations! Your manuscript is now being handed over to our production team.

Kind regards,

on behalf of

Dr. Ahmed M. Yosri

Academic Editor

PLOS ONE